# Dynamical Majorana edge modes in a broad class of topological mechanical systems

Emil Prodan[1], Kyle Dobiszewski[2], Alokik Kanwal[2], John Palmieri[3] & Camelia Prodan[2]

Mechanical systems can display topological characteristics similar to that of topological insulators. Here we report a large class of topological mechanical systems related to the BDI symmetry class. These are self-assembled chains of rigid bodies with an inversion centre and no reflection planes. The particle-hole symmetry characteristic to the BDI symmetry class stems from the distinct behaviour of the translational and rotational degrees of freedom under inversion. This and other generic properties led us to the remarkable conclusion that, by adjusting the gyration radius of the bodies, one can always simultaneously open a gap in the phonon spectrum, lock-in all the characteristic symmetries and generate a non-trivial topological invariant. The particle-hole symmetry occurs around a finite frequency, and hence we can witness a dynamical topological Majorana edge mode. Contrasting a floppy mode occurring at zero frequency, a dynamical edge mode can absorb and store mechanical energy, potentially opening new applications of topological mechanics.

[1] Department of Physics, Yeshiva University, New York, New York 10016, USA. [2] Department of Physics, New Jersey Institute of Technology, Newark, New Jersey 07102, USA. [3] Department of Biomedical Engineering, New Jersey Institute of Technology, Newark, New Jersey 07102, USA. Correspondence and requests for materials should be addressed to E.P. (email: prodan@yu.edu).

Topological mechanics evolved in an extremely vigorous field and, in a relatively short time span, we have seen theoretical and experimental realizations of such topological effects in purely mechanical settings[1–23], something that would have been hard to imagine just a few years ago. Among the most striking examples are the mechanical analogues of the quantum Hall and spin-Hall effects where, for example, the latter requires a half-integer spin that has no direct equivalent in the classical realm. Here we report another intriguing correspondence between quantum and classical mechanics, realized by mechanical analogues of band Hamiltonians describing the fermionic excitations of one-dimensional (1D) topological superconductors from the BDI symmetry class[24]. They are linear periodic chains of rigid bodies with an inversion symmetry point, where the particle-hole symmetry stems from the distinct behaviour of the translational and rotational degrees of freedom under the inversion operation and it manifests as a mirror symmetry of the phonon spectrum. An interesting finding is that the particle-hole symmetry occurs relative to a finite energy and, as a consequence, the topological Majorana edge modes exist at a finite frequency. Hence, they have non-trivial dynamics, in particular, they can collect and store energy that can enable a completely new array of applications.

A few comments about the terminology are in order. In the context of fermionic 1D topological superconductors from the BDI symmetry class[24], the topologically protected edge excitations are termed Majorana fermions[25]. This is because these states are their self-mirror image under the charge conjugation, very much like the fermion proposed by Majorana[26] is its self anti-particle. In this work we propose the same terminology for the edge modes of our systems, but with the word fermion crossed out because the statistics does not play any role here. We believe the terminology is appropriate and useful because the edge states are invariant under a charge conjugation. This gives them a distinct feature that separates them from other ordinary vibrational edge modes that can appear by accident. For example, the motion of the Majorana mode remains unchanged if we exchange the rotational and translational degrees of freedom and follow with time-reversal operation. Furthermore, the characteristic symmetries of the BDI class are locked-in for these mechanical systems, since the only way to violate them is to destroy the rigid bodies making up the system. In the other context, this will be equivalent to destroying the superconducting phase.

Our study goes beyond the simple reporting of a topological mechanical system from the BDI symmetry class and, in fact, it is focused on much broader issues as we now explain. Let us first introduce the context. Guided by the protein structure of microtubules and motivated by their yet not fully understood dynamical instability[27], Prodan and Prodan[1] designed a topological Chern phononic crystal and put forward the thesis that its topological vibrational edge modes may assist in the destabilization of the microtubules' caps that triggers the depolymerization process[28]. Along similar lines, Berg et al.[2] generated a mechanical model of the actin filaments displaying topological edge modes, a finding that could support/explain the brownian ratchet hypothesis[29], central to understanding how actin filaments push against the cell membrane[30]. The proposed topological phonon-assisted mechanisms were described in detail by our previous works but the following pressing questions need to be answered before such proposals can be taken seriously: How large can a class of such topological structures be within the generic class of self-assembled structures? What kind of fine-tuning is required for a mechanical structure to enter a topological phase? Our study substantiates these issues by reporting a large class of mechanical systems that can be always

tuned into a topological phase using a single parameter that, in practice, can be as simple as modifying a gyration radius. More precisely, we show that, upon varying this parameter, the following required conditions are simultaneously satisfied: first, the bulk phonon spectrum has a gap; second, the required BDI symmetries are all in place and locked, even when a boundary is present; and third, the bulk topological invariant is non-trivial. Let us clarify that we need to tune the parameter into a single value rather than an interval. Still, the remarkable fact here is that, typically, enforcing all these three conditions simultaneously will require a three-dimensional (3D) tuning space, that is, one dimension per constraint, yet here we can achieve all of the above in a 1D tuning space. The latter is special because, when continuously evolving in the right direction, the system will necessarily cross the desired value but this is not at all the case if the tuning space was of higher dimension. As such, the tuning may even happen by random chance, as often is the case in nature, hence one could wonder whether the living organisms have indeed stumbled upon such exotic structures during the millions of years of evolution.

## Results

**A concrete example**. We start our analysis with a concrete example that requires no tuning at all. It consists of a periodic array of dimers interconnected by two springs, as described in Fig. 1. The small oscillations of the system are analysed theoretically in Supplementary Note 1. When the two springs are identical as in Fig. 1a, the system has inversion and mirror symmetries and displays ungapped phonon spectrum (see Fig. 1b). As soon as the springs are not identical as in Fig. 1d, the mirror symmetry is lost and the system displays a gapped spectrum (see Fig. 1e). Note that the inversion symmetry remains unbroken and this leads to the particle-hole symmetry mentioned above that becomes visible once we plot the pulsations squared (see Fig. 1c,f). The particle-hole symmetry together with the intrinsic time-reversal symmetry provide the characteristic symmetries of the band Hamiltonians from the BDI symmetry class. For this model, one can straightforwardly evaluate the standard topological invariant[31] associated with the BDI symmetry class (see equation (15)), and the result is always $\pm 1$ and never zero. Since closing the spectral gap requires fine-tuning, we can rightfully say that this particular class of dimer chains always exists in a gapped topological phase. A laboratory realization of the system is reported in Fig. 2 and the expected Majorana edge modes are documented in Fig. 2b. All experimental parameters are supplied in the Supplementary Fig. 1. The dynamics of the system and of the dynamical Majorana edge modes can be further analysed in Supplementary Movies 1 and 2, where the chirality of the modes can be clearly identified. Let us point out that the system requires no fine-tuning precisely because of the particular connection of the springs. For example, if the springs were not connected to the centres of the spheres but somewhere on the connecting rod, then a tuning would be necessary, as further discussed below. The present experimental model is similar yet quite different from the system analysed theoretically in Berg et al.[2]. There, the presence of two additional springs can break the particle-hole symmetry and, as a consequence, the system was analysed in Berg et al.[2] solely from the point of view of an inversion-symmetric system, a class known to accept non-trivial topological classification[32,33]. For this reason, the topological invariant invoked in Berg et al.[2] is of very different nature from the standard invariant of the BDI symmetry class used here. Note, however, that the particle-hole symmetry of the chain analysed in Berg et al.[2] can be always restored using the tuning mechanism introduced below.

**Analysis of the general case.** In the following, we show that the experimental model is actually part of the vastly larger topological class depicted in Fig. 3. Key assumptions for our analysis are the inversion symmetry of the dimers, lack of any reflection symmetry planes and the existence of only two relevant degrees of freedom per dimer, $\mathbf{x} = (x_t, x_r)$, of which one is translational and one is rotational (that is, an angle), as in Fig. 3b. When we do not need to emphasize the distinct nature of the degrees of freedom, we will use the more convenient notation $\mathbf{x} = (x_1, x_2)$. We will multiply the angles by the radius of gyration $d = \sqrt{I/M}$, such that both degrees of freedom have the unit of length and the kinetic energy is isotropic. Here, $I$ and $M$ are the moment of inertia relative to the inversion point and the mass of the rigid bodies, respectively. The equilibrium configuration of the chain is assumed linear and periodic as in Fig. 3a that, among other things, ensures that the whole structure is symmetric relative to inversion.

The rigid bodies interact pairwise, and hence the dynamics of the chain is determined by a generic Lagrangian of the form:

$$\mathcal{L} = \frac{1}{2} M \sum_{n \in \mathbb{Z}} \left( \dot{x}_t(n)^2 + \dot{x}_r(n)^2 \right) - \sum_{q>0} \sum_{n \in \mathbb{Z}} V_q(\mathbf{x}(n), \mathbf{x}(n+q)) \quad (1)$$

Note that $q$, that labels the rank of the neighbours, assumes only positive values in equation (1). Later, we will let $q$ assume negative values too, in which case $V_{-q}(\mathbf{x}, \mathbf{x}') = V_q(\mathbf{x}', \mathbf{x})$. The linearized Lagrange equations take the form:

$$M \ddot{\mathbf{x}}(n, t) = -\sum_{q \in \mathbb{Z}} \widehat{Q}_q \mathbf{x}(n+q, t), \quad n \in \mathbb{Z} \quad (2)$$

where $\widehat{Q}_q$ are the $2 \times 2$ matrices:

$$\widehat{Q}_0(i,j) = \sum_{q \neq 0} \frac{\partial^2 V_q(\mathbf{x}, \mathbf{x}')}{\partial x_i \partial x_j}, \quad \widehat{Q}_q(i,j) = \frac{\partial^2 V_q(\mathbf{x}, \mathbf{x}')}{\partial x_i \partial x'_j}, \quad (3)$$

$$q \neq 0$$

with the derivatives taken at the equilibrium configuration. The matrices are real and $\widehat{Q}_{-q}$ equals the transpose of $\widehat{Q}_q$. Throughout, we will consistently indicate the matrices by a hat and the vectors by bold symbols. Passing to the frequency-momentum domain, $\mathbf{x}(n,t) = \mathbf{A}(k, \omega) e^{i(\omega t - kn)}$, we obtain the dispersion equation:

$$\omega^2 M \mathbf{A}(k, \omega) = \widehat{D}(k) \mathbf{A}(k, \omega) \quad (4)$$

with

$$\widehat{D}(k) = \sum_q \left[ \left( \widehat{Q}_q + \widehat{Q}_{-q} \right) \cos(qk) + i \left( \widehat{Q}_q - \widehat{Q}_{-q} \right) \sin(qk) \right] \quad (5)$$

This dynamical matrix is a positive $2 \times 2$ Hermitean matrix with $\widehat{D}(k)^* = \widehat{D}(-k)$ that reflects the time-reversal symmetry of the dynamics.

The dynamical matrix is said to possess the chiral symmetry if, after removing an uninteresting constant diagonal part, the matrix can be sent into minus itself using a symmetry $\widehat{S}$:

$$\widehat{S} \left( \widehat{D}(k) - \mu \widehat{I}_{2 \times 2} \right) \widehat{S}^{-1} = - \left( \widehat{D}(k) - \mu \widehat{I}_{2 \times 2} \right), \quad \widehat{S}^{\dagger} = \widehat{S},$$
$$\widehat{S}^2 = \widehat{I}_{2 \times 2} \quad (6)$$

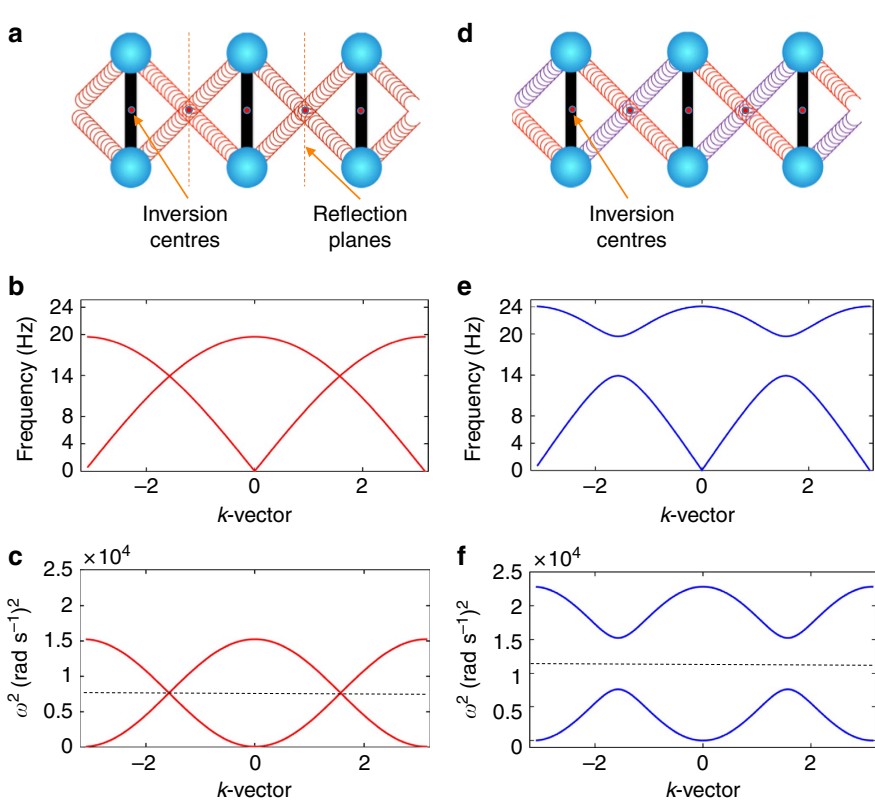

**Figure 1 | Example of a dimer chain that posses the particle-hole symmetry without any fine-tuning.** The dimers can rotate in the plane and the centre of the dimers can move along the longitudinal axis of the chain. The dimers are interconnected by two springs attached directly to the point masses. In (**a**), the two springs are identical and the system has both the inversion and reflection symmetries relative to the marked inversion centres and reflection planes, respectively. In this case, the phonon spectrum, shown (**b**), is ungapped. In (**d**), the two springs are different and only the inversion symmetry remains. In this case, the phonon spectrum, shown in (**e**), is gapped and the chain is in a topological phase from the BDI symmetry class, with bulk winding number $\nu = 1$. If the frequency is replaced by $\omega^2$ (pulsation squared), then the particle-hole symmetry becomes explicit in (**c,f**), relative to the dotted lines. The phonon spectrum in (**e**) was computed with theoretical parameters that fit the experimental phonon spectrum.

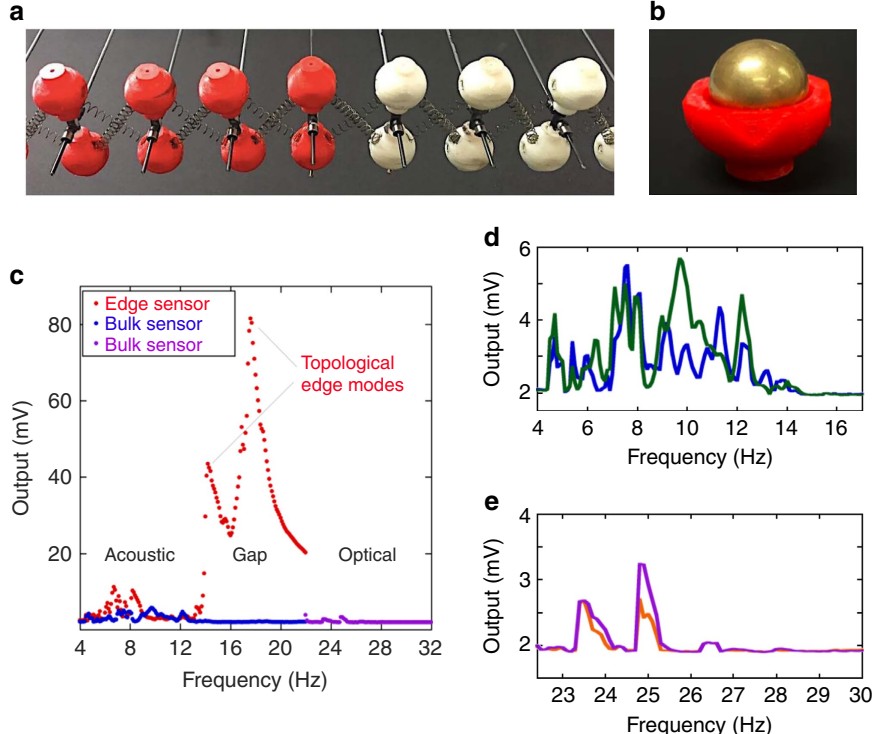

**Figure 2 | Experimental measurements.** (**a**) The experimental model consists of 32 red and 9 white dimers connected via springs. There is one such spring on one diagonal, while on the other there are two of them, intertwined. The configuration of the springs is switched between the red and white dimers, so that the red and white chains are inverted images of each other. This assures that the two chains are in topological phases with opposite bulk invariants. Hence, edge modes are expected at the boundary of the two phases. (**b**) Each dimer is made of two brass balls encapsulated in 3D printed plastic shells and connected by a plastic rod. (**c**) Typical measurements of the phonon response of the system. Each data point represents the root mean square of the output voltage of an accelerometer sensor attached to a ball and aligned with the axis of the chain. The red and blue curves represent simultaneous recordings from two sensors, one placed at the boundary and one in the bulk, respectively, while the actuator was attached to the red boundary dimer. The purple curve represents the recording from a sensor placed in the bulk, while the actuator was attached to the left free edge of the red chain. The phonon gap is clearly visible and the two expected topological edge modes appear prominently in the gap. (**d**) Simultaneous recordings from two sensors placed on different dimers in the bulk, while the actuator was attached to a boundary dimer. The blue line is identical with the one in (**c**). (**e**) Simultaneous recordings from two sensors placed on different dimers in the bulk, while the actuator is attached to the free left edge. The purple line is identical with one in (**c**).

We recall that chiral and time-reversal symmetries automatically imply the particle-hole symmetry. The first remarkable property of the proposed systems is that inversion symmetry alone induces a chiral symmetry on the traceless part of $\widehat{D}(k)$. Indeed, as shown in Fig. 3b, under the inversion relative to the centre of a dimer at equilibrium, the translational degrees of freedom transform as $x_t(n) \to -x_t(-n)$, while the rotational ones transform as $x_r(n) \to x_r(-n)$. The last rule reflects the universal fact that, in two dimensions, lines remain parallel when inverted relative to any point in space. As such, the inversion operation takes the form $\mathbf{x}(n) \to \widehat{\sigma}_3 \mathbf{x}(-n)$ and the inversion symmetry of the original Lagrangian implies:

$$\widehat{D}(k) = \widehat{\sigma}_3 \widehat{D}(-k)\widehat{\sigma}_3 \qquad (7)$$

Examining the structure in equation (5), we see that $\widehat{Q}_q + \widehat{Q}_{-q}$ must commute while $i(\widehat{Q}_q - \widehat{Q}_{-q})$ anti-commutes with $\widehat{\sigma}_3$. Furthermore, $i(\widehat{Q}_q - \widehat{Q}_{-q})$ is traceless, self-adjoint and purely imaginary. Hence, it must be proportional with $\widehat{\sigma}_2$. In other words:

$$\widehat{D}(k) = \mu(k)\widehat{I}_{2\times2} + d_2(k)\widehat{\sigma}_2 + d_3(k)\widehat{\sigma}_3$$
$$\equiv \mu(k)\widehat{I}_{2\times2} + \widehat{\mathcal{D}}(k) \qquad (8)$$

We now can see explicitly that the traceless part of the dynamical matrix has indeed the chiral symmetry:

$$\widehat{\sigma}_1 \widehat{\mathcal{D}}(k)\widehat{\sigma}_1 = -\widehat{\mathcal{D}}(k) \qquad (9)$$

Remarkably, the chiral symmetry, implemented by $\widehat{\sigma}_1$, interchanges the rotational and translational degrees of freedom. Let us point out that it is the traceless part $\widehat{\mathcal{D}}(k)$ that determines the direct gap in the phonon spectrum, while the diagonal term $\mu(k)\widehat{I}_{2\times2}$ shifts the bands by an equal amount. The bulk invariant from equation (15) can be already defined at this level and a bulk classification can be given in the sense that two chains with different bulk topological invariants cannot be adiabatically deformed into each other without closing the bulk direct gap. However, the bulk-boundary correspondence is not present unless $\mu(k)$ is a pure constant[31].

We now restrict to the typical case of nearest-neighbour interactions $(q = 1)$ and we assume that the radius of gyration $d$ can be adjusted without modifying the interaction potential. This will be used next to set $\mu(k) = \mathrm{Tr}(\widehat{D}(k))$ in equation (8) to a true constant. For this we need to examine some particularities of the system. The first observation is that mechanical stability requires:

$$Q_0(i,i) > 0, \quad i = 1, 2 \qquad (10)$$

and, since the pair-interactions depend only on $x_t(n) - x_t(n \pm 1)$, it follows from equation (3) that:

$$\widehat{Q}_{-1}(1,1) = \widehat{Q}_{+1}(1,1) = -\tfrac{1}{2}\widehat{Q}_0(1,1) \qquad (11)$$

hence $\widehat{Q}_{\pm1}(1,1)$ are necessarily negative. Equation (11) also warrants the existence of an acoustic dispersion band touching the zero frequency at $k = 0$. Regarding the response of the system

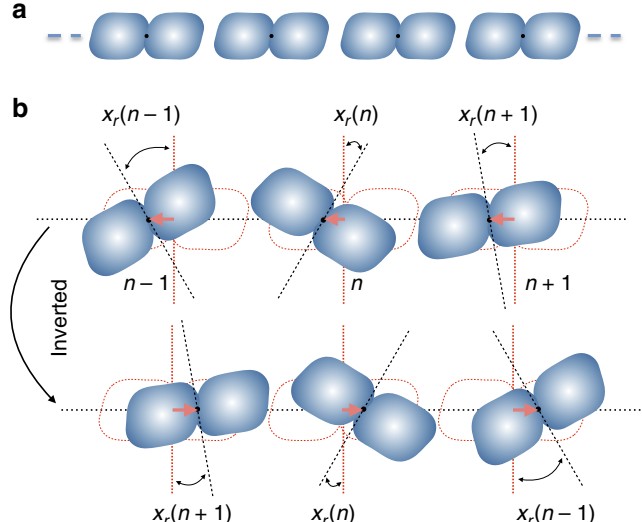

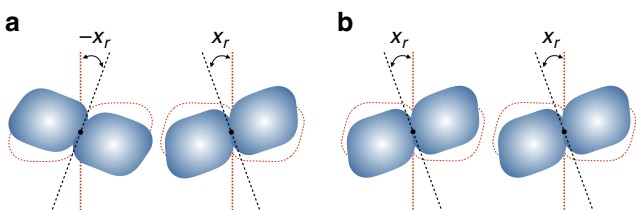

**Figure 3 | The generic topological system.** (**a**) The equilibrium configuration of a self-assembled periodic array of rigid dimers with inversion symmetry relative to their centre of mass (marked by the dot). One important observation is that, regardless of the details of the self-assembly process, the periodic array, as a whole, has the inversion symmetry. (**b**) A dynamical configuration of the dimer chain showing the relevant degrees of freedom that are the translations of the dimer centres along the axis of the chain and the rotation of the dimers. It also shows how the degrees of freedom transform under the inversion operation. The key observation here is that the translational degrees of freedom change sign while the rotational ones do not.

**Figure 4 | Behaviour under different rotations.** Rotation of the dimers by opposite angles (**a**) corresponds to bending the dimer chain, while a rotation by identical angles (**b**) is closely related to shearing the dimer chain.

to rotations of the dimers, we point to the universal facts that chains are easy to bend yet difficult to shear apart. Hence, the restoring forces are generally weak when the dimers are rotated by opposite angles as in Fig. 4a and are strong when the dimers are rotated by equal angles as in Fig. 4b. In other words, the pair potentials $V_{\pm 1}$ are primarily functions of the combinations $x_r(n) + x_r(n \pm 1)$ and, as a consequence:

$$Q_{-1}(2,2) = Q_{+1}(2,2) \approx \tfrac{1}{2}Q_0(2,2) \qquad (12)$$

In particular, $Q_{\pm 1}(2,2)$ are positive. These features, which follow directly from the very nature of the dimer chains and are not a result of some fine-tuning, lead to a string of remarkable consequences.

The first consequence is that the coefficient

$$\mu(k) = \frac{1}{2}\left[\widehat{Q}_0(1,1) + \widehat{Q}_0(2,2)\right] + \left[\widehat{Q}_1(1,1) + \widehat{Q}_1(2,2)\right]\cos(k) \qquad (13)$$

can be set to a constant by adjusting the parameter $d$. Indeed, the positive term $\widehat{Q}_1(2,2)$ scales as $d^{-2}$, while the negative term

$\widehat{Q}_1(1,1)$ does not have such dependence. Hence, exact cancellation in the second term of equation (13) can be always achieved by varying $d$. In this case, the band spectrum of $\omega^2$ will display the particle-hole symmetry.

The second consequence is that a full spectral gap is generically opened in the phonon spectrum. Indeed, note that if the system possesses an additional reflection symmetry (as in Fig. 1a), then the term involving $d_3(k)$ is missing in equation (8), and the two spectral bands of $\widehat{D}(k)$ cross at $k = \frac{\pi}{2}$ as in Fig. 1b. Furthermore, due to the characteristics discussed above, the bands associated with the translational and rotational degrees of freedom have a minimum at $k = 0$ and $k = \pi$, and a maximum at $k = \pi$ and $k = 0$, respectively. This leads to the particular crossing of the bands seen in Fig. 1b so that when the degeneracy at $k = \frac{\pi}{2}$ is lifted by breaking the reflection symmetry, which was our original assumption, the bands split and a full spectral gap opens in the phonon spectrum as in Fig. 1e. Note that this will not be the case if the bands cross, for example, as in Fig. 6a of Berg et al.[2]

The third consequence is that the topological invariant is always non-trivial. Indeed, let us represent Pauli's matrices in the basis $\frac{1}{\sqrt{2}}\begin{pmatrix} 1 \\ 1 \end{pmatrix}$ and $\frac{1}{\sqrt{2}}\begin{pmatrix} 1 \\ -1 \end{pmatrix}$ so that $\widehat{\sigma}_1$ is diagonal $\widehat{\sigma}_1 = \begin{pmatrix} 1 & 0 \\ 0 & -1 \end{pmatrix}$. In this case:

$$\widehat{\mathcal{D}}(k) = \begin{pmatrix} 0 & f(k)^* \\ f(k) & 0 \end{pmatrix}, \quad f(k) = d_3(k) + id_2(k) \qquad (14)$$

and it is easy to see that the phonon spectrum is composed of two bands separated by a direct spectral gap $\Delta(k) = 2|f(k)|$. As long as this direct gap remains open, which we already know is generically the case, the winding number:

$$\nu = \int_0^{2\pi} \frac{\mathrm{d}f(k)}{f(k)} \qquad (15)$$

is a well-defined quantized topological invariant[31]. Geometrically, $\nu$ counts the windings around the origin of the loop traced by $f(k)$ in the complex plane as $k$ is varied from 0 to $2\pi$. Explicitly:

$$f(k) = \frac{1}{2}\mathrm{Tr}\left\{\widehat{\sigma}_3\widehat{Q}_0\right\} + \frac{1}{2}\mathrm{Tr}\left\{\widehat{\sigma}_3\left(\widehat{Q}_1 + \widehat{Q}_{-1}\right)\right\}\cos(k) \\ + i\,\mathrm{Tr}\left\{\widehat{\sigma}_2\left(\widehat{Q}_1 + \widehat{Q}_{-1}\right)\right\}\sin(k) \qquad (16)$$

that is the equation of an ellipse in the complex plane, centred at $x_c = \frac{1}{2}\left[\widehat{Q}_0(1,1) - \widehat{Q}_0(2,2)\right]$ and of horizontal semi-axis $b = \frac{1}{2}\left|\widehat{Q}_1(1,1) - \widehat{Q}_1(2,2)\right|$. The winding number is $\pm 1$ if the origin of the coordinate system is located inside this ellipse (that is, $b > x_c$) and zero otherwise. As a consequence, we can write a simple condition for a non-trivial topological number $\nu \neq 0$:

$$\left|\widehat{Q}_1(2,2) - \widehat{Q}_1(1,1)\right| > \frac{1}{2}\left|\widehat{Q}_0(2,2) - \widehat{Q}_0(1,1)\right| \qquad (17)$$

However, this condition is obviously satisfied because, if we add equations (11 and 12) we obtain:

$$Q_1(2,2) - Q_1(1,1) \approx \frac{1}{2}\left(Q_0(2,2) + Q_0(1,1)\right) \qquad (18)$$

that, together with the positive character of $Q_0(i,i)$, automatically implies equation (17).

**Analysis of the edge modes.** When an edge is present, the bulk-boundary correspondence for the BDI symmetry class in one-dimension reads[34]:

$$\nu = N_+ - N_- \qquad (19)$$

where $N_{\pm}$ are the number of topological Majorana edge modes of positive/negative chirality. Note that this principle ensures the existence of at least $v$ Majorana edge modes that in our case occur in the middle of the bulk spectral gap at a finite frequency. The key condition for equation (19) to hold is that the edge does not break the particle-hole symmetry of the system. Another remarkable fact is that this is indeed the case when we sever the dimer chain in half because, with the tuning from point (1), all three matrices $\widehat{Q}_0 - \mu I_{2\times2}$ and $\widehat{Q}_{\pm1}$ anti-commute with $\sigma_1$. In other words, the normal-mode equation in the presence of an edge

$$
\begin{aligned}
\left(\omega^2 M - \mu\right)\mathbf{x}(n,\omega) =& \widehat{Q}_{-1}\mathbf{x}(n-1,\omega) + (\widehat{Q}_0 - \mu I_{2\times2})\mathbf{x}(n,\omega) \\
&+ \widehat{Q}_{+1}\mathbf{x}(n+1,\omega)
\end{aligned}
$$

$$(20)$$

with $n > 0$ and $\mathbf{x}(0,\ \omega) = 0$ continues to display the chiral symmetry with respect to $\widehat{\sigma}_1$. This is precisely the setting for the bulk and boundary index theorems formulated in ref. 31, leading to a rigorous proof of equation (19), and hence the existence of robust Majorana edge modes is proven. For the same reason, the chiral symmetry remains unbroken when two chains with opposite topological numbers are connected as in the experiment of Fig. 2, in which case the bulk-boundary correspondence principle predicts two Majorana edge modes as it was indeed observed. The splitting of the modes from the predicted mid-gap position is attributed to non-linear effects and to a small possible symmetry breaking.

## Discussion

The main message of our work is that, if particles with an inversion centre, such as dimers, self-assemble into periodic linear chains, then by a simple one-parameter tuning the chains can be driven into a topological phase from the BDI symmetry class. Let us point out that the building blocks of many soft materials in living organisms are protein dimers and such self-assembled structures are common[35]. It was this observation that prompted us in the first place to focus on this type of systems. In the real world, of course, the systems are much more complex and will certainly have additional degrees of freedom. Note however, that our hypothesis was about the relevant degrees of freedom, namely, those that, to a high degree, determine the phonon spectrum around the topological gap. For example, our experimental model certainly had additional degrees of freedom. The centre of mass of the dimers was not constrained on a line and the whole chain was actually very soft against bending. For this very reason, the bending modes died out at very low frequencies, and therefore we could ignore them entirely at frequencies around the topological gap.

The tuning we propose is through the gyration radius $d = \sqrt{I/M}$ that can be easily changed by adding/subtracting mass at/from near the inversion point, hence leaving $I$ virtually the same, or by redistributing the mass and hence changing $I$. Both actions can be in principle accomplished without substantial changes in the interaction between the rigid bodies. At the microscale, the gyration radius can be varied, for example, by exploiting the thermal expansion of the materials or, for soft matter, by manipulating the medium conditions to induce changes in the conformation of the dimers. The tuning can also be achieved by varying the interactions, like in the case of crystalline structures made with nanoscale programmable atom equivalents (gold nanoparticles with a dense layer of DNA), where plasticity can be changed on demand by modifying the chemical cues of the buffer[36].

To summarize, we found that self-assembled periodic arrays of inversion-symmetric particles can acquire a particle-hole symmetry under a mild tuning, in which case they generically enter a topological mechanical phase where dynamical Majorana edge modes are observed. Key features of the dynamical Majorana modes are: (1) their symmetry with respect to the charge conjugation, implemented by the exchange of the rotational and translational degrees of freedom followed by time reversal, (2) their stable positioning in the middle of the bulk phonon gap and (3) the capacity to absorb and store energy without spilling it into the bulk of the chain, a phenomenon that can be clearly observed in the Supplementary Movies 1 and 2. Perhaps the class of topological self-assembled chains discovered here is just one among many others that remain to be discovered but, in our opinion, their existence already provides more support for the thesis that topological mechanical systems can occur naturally outside laboratories. As opposed to the gyroscope-assisted lattices[18], the topology in the systems studied here emerges solely from their structure. As such, new materials with dynamic topological edge modes can be already engineered at micron and nanoscale using the recent advances in self or directed assembled materials. Indeed, methods to design and self-assemble nano-structures with programmable lattices are widely available and, for example, DNA and proteins, such as clathrin, have been used extensively to create 3D lattices or other solid state-like structures, 3D origami and even origami lattices[37,38].

## Methods

**Assembly of the mechanical system.** The dimers were produced by bonding plastic half-shells of 1.15 inches in outer diameter around brass spheres of diameter 0.75 inches, as illustrated in Fig. 2b. The half-shells were produced of acrylonitrile butadiene styrene through fused deposition modelling with a Hyrel 3D System 30M printer (Hyrel, Norcross, GA, USA). The plastic half-shells were designed with indentations at various points to facilitate proper alignment of the hemi-spheres and attachment of springs and rods at precise angles. A custom-designed alignment jig facilitated the precise and uniform assembly of the dimer units. To form the dimers, an acetal rod of 0.25 inch in diameter and 1 inch in length was inserted into the indentations on two individual plastic/brass spheres. Acetal was chosen because of its low coefficient of friction, and this was important because the system is suspended from the centre of mass of the dimers that is in the middle of the rods. Another custom alignment jig was utilized to ensure the precise and uniform connection of the dimers. Springs of 50.8 mm length and 200 N m$^{-1}$ spring constant (supplied by Associate Spring Raymond) were affixed to the dimers using the angled indentations on the plastic/brass spheres, as illustrated in Fig. 2a. To suspend the system, 3/32 inch through holes were drilled vertically in the centre of the acetal rod connecting each dimer, which was facilitated again by the use of a custom drilling jig, and then a 3/32 inch chucking reamer was utilized to further process each hole. A custom ball bearing system was designed to suspend the system and allow nearly frictionless motion of the system. For this, 3/32 inch diameter steel rods were inserted through the vertical through holes in the acetal rod. Metalized (silver) plastic beads (3 mm in diameter) were utilized to 'sandwich' the acetal rod and the beads were held in place by steel collar nuts. To further reduce the friction, a graphite dry lubricant (Blaster Corporation Valley View, OH, USA) was applied to both the steel and acetal rods. Small ball links were utilized to attach the vertical steel rods to a horizontal, stationary aluminum beam at precise intervals. All the hardware utilized in the suspension of the model system was purchased from Du-Bro Products, Inc. (Wauconda, IL, USA).

**Data recording and analysis.** The experiment was run using a custom-built computer-controlled system coordinated by LabVIEW. A sinusoidal signal of varying frequencies was produced using an arbitrary waveform generator (HP 33120A) connected to a computer via a GPIB interface. The sinusoidal signal was amplified by a Pasco function generator (TI8127) that was set up to accept an external signal. The amplified signal was sent to a string vibrator (Pasco WA9857) that was attached to a dimer via a screw. The kinematics of the dimers was captured using analogue accelerometers and their evaluation boards (eval-adxl326z). The accelerometers produced two voltage signals proportional to the accelerations parallel and perpendicular to the chain's axis. The signals were captured using a USB data acquisition system (NI USB6216).

The LabVIEW program sets the frequency and outputs the sinusoidal signal on the actuator. The actuator is set in motion and the system is allowed for 5 s to reach its stationary regime and afterwards the data from the accelerometers is captured for 10 s. Then, the sinusoidal signal is turned off and the system is allowed to come to rest for 5 s. The frequency is increased (in steps of 0.1 Hz) and the cycle starts again. The result is 100,000 measurements per frequency and, for each frequency, the LabVIEW program calculates and subtracts the base line and then calculates

the root mean square (r.m.s.) of the calibrated data. The r.m.s. is proportional with the amplitude of the acceleration at that particular frequency. The r.m.s. values are plotted in Fig. 2. The frequency interval combed during a whole experiment was 0–35 Hz.

**Data availability.** All relevant data are available upon request from the authors.

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

## Acknowledgements

E.P. acknowledges support from the Keck Foundation and NSF through the grant DMR-1056168. C.P. acknowledges support from the Keck Foundation and NSF I-Corps Site at NJIT. J.P. acknowledges support from NJIT Provost Research Program.

## Author contributions

E.P. performed the theoretical analysis. K.D., A.K., J.P. and C.P. participated equally in the experimental design, data acquisition and data analysis. All authors contributed to the typesetting of the manuscript.

## Additional information

**Competing financial interests:** The authors declare no competing financial interests.

**Publisher's note**: 

