## [Peer Review File · Nature Communications]

Reviewers' comments:

Reviewer #1 (Remarks to the Author):

The authors study the topological properties of mechanical excitations in periodic dimer chains coupled by harmonic springs. They show that when the dimer chains satisfy certain conditions (inversion symmetry, lack of reflection symmetry, two relevant degrees of freedom), the band structure of excitations is gapped and has a particle-hole symmetry which puts the system in the BDI topological class with nontrivial index by tuning only one parameter (the radius of gyration of the dimers). Mid-gap chiral edge modes are predicted and observed in an experimental realization using plastic and metal parts. Because a series of conditions can be simultaneously satisfied by tuning a single parameter, the authors claim that experimental systems can easily be brought into the regime with topological edge modes, and furthermore that such structures could potentially exist in nature.

The revision successfully addresses many of the concerns raised by the present reviewer (Reviewer #1) about the initial submission. The presentation in terms of the properly defined parameter 'd' is easier to follow than the earlier two-step tuning process, and makes the argument of easy tunability more convincing. The statements about the 'ubiquitous' nature of such assemblies has also been moderated in the revision, and the paper does support the more modest claims made in the abstract and conclusions. The paper is of interest to the topological mechanical community. I think it would require a more serious analysis of a real biological or colloidal system to reach out to the broader soft matter/biophysics audience, but I support its publication as an important first step identifying a new class of topological mechanical systems.

I have one remaining scientific point which I would like to see addressed: the authors repeatedly talk about the inversion symmetry of the individual dimers, but it seems that the assembly as a whole also needs to preserve the inversion symmetry, which is not guaranteed if the building blocks satisfy the symmetry e.g. dimers need to be in a perfectly straight line with equal spacing. It would be appropriate to also mention this condition among the assumptions needed for the reported phenomena to be observed.

In addition, I have the following editing comments/suggestions:

- the term "consecrated name" does not seem appropriate in a scientific context. I would edit the sentence to "In the context of fermionic 1-dimensional BDI topological superconductors, the topologically protected edge excitations are termed Majorana fermions." or similar.
- spelling errors/typos: simultaneously (line 26), Majorana (line 77), brake (111), 'is of very different' (114), guaranteed (172), cancelation (192), sheering (Fig 4 caption), 'for every red dimers' (SI last sentence). Not a comprehensive list.

Reviewer #2 (Remarks to the Author):

The revised version of the manuscript is very much improved and addresses most of the comments raised by the reviewers.

I find the discussion of the relation between the mechanical systems discussed in the present manuscript and the well known topological superconductors convincing, apart from the claim of the built-in symmetries. With the radius of gyration, a continuous parameter needs to be fine-tuned to achieve these symmetries which in my opinion is still much less generic than the occurrence of particle hole symmetry in superconductors. The issue of fine-tuning is not properly reflected yet in the prominent claim "which can be always tuned into a topological phase using a single parameter" in the introduction. It should be made clear that only for a fine-tuned value of this parameter the phase is defined, and not in a whole range. This point is also misleading in the authors' reply to reviewer 3, where they compare the role of their tuning parameter to the thickness of HgTe quantum wells in quantum spin Hall structures. There, the topological phase of the system changes at a critical value of the thickness and stays topologically non-trivial above this value. The time-reversal symmetry (relevant for the definition of the quantum spin Hall phase) is not affected at all by the thickness. In contrast, in the present work, only at an isolated value of the tuning parameter all symmetries necessary to enter the BDI class are satisfied. This distinction should be made clear.

Once the above point has been addressed in the manuscript, I recommend publication in Nature Communications, even though the fine-tuning issue somewhat weakens the case of this interesting manuscript.

On a more detailed note, there is a typo on page 4: "Mojorana" -> "Majorana".

Reviewer #3 (Remarks to the Author):

It's good that now the authors correct the statement on the "fine-tuning" aspects of the theory. The manuscript is certainly worth publication somewhere. However, I found the manuscript not containing enough new physics for the broad audience of Nature Communications. The type of 1D chain model has already been discussed by the authors in a number of previous papers. The current manuscript, which claims to make it more general, seems to be rather incremental.

REVIEWERS' COMMENTS:

Reviewer #1 (Remarks to the Author):

The revision successfully addresses my scientific concerns and I recommend publication.

Reviewer #2 (Remarks to the Author):

The authors have addressed my final comments and also reply convincingly to the other Reviewers. The scope and motivation of this interesting study is now very clearly presented. I fully recommend publication of the manuscript in its present form.

Dear Editors

In this letter we present our response to the final comments of the reviewers and the changes made to text.

Reviewer 1:

"I have one remaining scientific point which I would like to see addressed: the authors repeatedly talk about the inversion symmetry of the individual dimers, but it seems that the assembly as a whole also needs to preserve the inversion symmetry, which is not guaranteed if the building blocks satisfy the symmetry e.g. dimers need to be in a perfectly straight line with equal spacing. It would be appropriate to also mention this condition among the assumptions needed for the reported phenomena to be observed."

We now make sure we always refer to the system as linear and periodic. The referee's remark about the inversion symmetry is explicitly addressed at lines 119-121, in the paragraph where we state all our assumptions.

"In addition, I have the following editing comments/suggestions:

- the term "consecrated name" does not seem appropriate in a scientific context. I would edit the sentence to "In the context of fermionic 1-dimensional BDI topological superconductors, the topologically protected edge excitations are termed Majorana fermions." or similar."

We have followed the recommendation and replace the text as suggested above.

"- spelling errors/typos: simultaneously (line 26), Majorana (line 77), brake (111), 'is of very different' (114), guaranteed (172), cancelation (192), sheering (Fig 4 caption), 'for every red dimers' (SI last sentence). Not a comprehensive list."

We thank Reviewer 1 for the careful reading. We have corrected these typos.

Reviewer 2

"I find the discussion of the relation between the mechanical systems discussed in the present manuscript and the well known topological superconductors convincing, apart from the claim of the built-in symmetries. With the radius of gyration, a continuous parameter needs to be fine-tuned to achieve these symmetries which in my opinion is still much less generic than the occurrence of particle hole symmetry in superconductors. The issue of fine-tuning is not properly reflected yet in the prominent claim "which can be always tuned into a topological phase using a single parameter" in the introduction. It should be made clear that only for a fine-tuned value of this parameter the phase is defined,

and not in a whole range.”

We now state explicitly at line 56 that one needs to be tuned into a single value rather than a whole interval. As before, we follow up by pointing out that this value can be always matched by a simple continuous deformation of the one parameter.

“On a more detailed note, there is a typo on page 4: "Mojorana" -> "Majorana".

We have corrected this typo, thank you.

Reviewer 3

“It's good that now the authors correct the statement on the "fine-tuning" aspects of the theory. The manuscript is certainly worth publication somewhere. However, I found the manuscript not containing enough new physics for the broad audience of Nature Communications. The type of 1D chain model has already been discussed by the authors in a number of previous papers. The current manuscript, which claims to make it more general, seems to be rather incremental.”

We thank the reviewer for his/her positive remarks and for pointing out that the new physics need to be highlighted. While indeed a somewhat similar system was analyzed in the literature, there was no mention of the particle-hole symmetry which stabilizes the Majorana edge modes. Let us stress point by point why we think our present findings are actually out of the ordinary:

1) At the first sight, one would be inclined to infer that the particle-hole symmetry (relative to a finite frequency!) can occur only in extremely specialized cases. What we found is quite the contrary: **the particle-hole symmetry always emerges in linear periodic chains with inversion symmetry upon the variation of the gyration radius of the rigid bodies**. Furthermore, the chiral symmetry is implemented by exchanging the translational and rotational degrees of freedom, in stark contrast to other examples where the chiral symmetry due to a spatial symmetry (the sublattice symmetry). This is new.

2) Even with this symmetry in place, one would be inclined to infer that the chances for a spectral gap will be minuscule. Quite the contrary, we found that a phonon spectral gap is generically opened. This is new

3) With the inversion and the spectral gap in place, one would be inclined to say that there are generically 50-50% chances for the system to be topological/non-topological. Quite the contrary, we find that these systems always exist in the topological phase. This is new.

This news is worth a wide dissemination to our colleagues in hard and soft-matter physics or mechanical engineers, who now can legitimately include the topological effects among their working hypotheses or research goals when researching engineered or self-assembled chains.